# Self-Powered and Flexible Triboelectric Sensors with Oblique Morphology towards Smart Swallowing Rehabilitation Monitoring System

**DOI:** 10.3390/ma15062240

**Published:** 2022-03-18

**Authors:** Jonghyeon Yun, Hyunwoo Cho, Jihyeon Park, Daewon Kim

**Affiliations:** 1Department of Electronics and Information Convergence Engineering, Kyung Hee University, 1732 Deogyeong-daero, Giheung-gu, Yongin 17104, Korea; jonghyeon.yun@khu.ac.kr (J.Y.); hyunwoo.cho@khu.ac.kr (H.C.); jihyeon.park@khu.ac.kr (J.P.); 2Institute for Wearable Convergence Electronics, Kyung Hee University, 1732 Deogyeong-daero, Giheung-gu, Yongin 17104, Korea; 3Department of Electronic Engineering, Kyung Hee University, 1732 Deogyeong-daero, Giheung-gu, Yongin 17104, Korea

**Keywords:** triboelectric nanogenerator, tilting reactive ion etching, smart swallowing rehabilitation

## Abstract

With aging, disability of the body can easily occur because the function of the body is degraded. Especially, swallowing disorder is regarded as a crucial issue because patients cannot obtain the nutrients from food by swallowing it. Hence, the rehabilitation of swallowing disorder is urgently required. However, the conventional device for swallowing rehabilitation has shown some limitations due to its external power source and internal circuit. Herein, a self-powered triboelectric nanogenerator for swallowing rehabilitation (TSR) is proposed. To increase the electrical output and pressure sensitivity of the TSR, the tilted reactive ion etching is conducted and the electrical output and pressure sensitivity are increased by 206% and 370%, respectively. The effect of the tilted reactive ion etching into the electrical output generated from the TSR is systematically analyzed. When the tongue is pressing, licking, and holding the TSR, each motion is successfully detected through the proposed TSR. Based on these results, the smart swallowing rehabilitation monitoring system (SSRMS) is implemented as the application and the SSRMS could successfully detect the pressing by the tongue. Considering these results, the SSRMS can be expected to be utilized as a promising smart swallowing rehabilitation monitoring system in near future.

## 1. Introduction

With the advancement of medical technology, increased income, and improved public health, the life expectancy of human beings has been gradually increased [1]. Accordingly, the importance of rehabilitation is increased because a person can suffer from a disability of the body for the rest of his or her life. Swallowing rehabilitation can be considered as the one of the most important rehabilitations to be performed. In order to maintain a healthy life, human beings should obtain the nutrients from the food and swallowing is the first process for nutrient absorption from food. If swallowing disorder occurs, the patient cannot properly swallow food, which can induce nutritional deficiencies. Moreover, the food can pass into the airways, resulting in sepsis and aspiration pneumonia. Conventionally, dysphagia (swallowing disorder) has been rehabilitated through diet [2,3] and exercise to increase the pressure of the tongue [4,5]. Among the two methodologies for swallowing rehabilitation, the exercises are a fundamental way to rehabilitate the swallowing function by increasing the pressure of the tongue [6,7,8]. The conventional device for increasing the pressure of the tongue is composed of modules for a detection and data process. Hence, the patient should hold the weight of both modules during the rehabilitation by their chin and teeth despite their poor strength. To process data, an external power source and internal circuit for charging are required for a conventional rehabilitation device, which increases the weight of the rehabilitation device. Hence, new devices for swallowing rehabilitation should have low weight and self-powered properties.

Energy harvesting technology [9,10,11,12,13] can be an attractive alternative to solve the issue related to conventional swallowing rehabilitation devices. This is because the mechanical energy is generated by the tongue during the rehabilitation activity and dissipated without any proper utilization. Hence, by adopting energy harvesting technology, this dissipated mechanical energy can be converted into the electrical energy, which can be utilized as the energy source of the device for rehabilitation. There are various types of energy harvesters that can convert mechanical energy into electrical energy, such as piezoelectric nanogenerators (PENG) [14,15,16,17,18], electromagnetic generators (EMG) [19,20,21], and triboelectric nanogenerators (TENG) [22,23,24,25,26]. Among them, the TENG has been considered as one of the most powerful solutions for new swallowing rehabilitation devices because the TENG generates triboelectricity from the friction between two different materials including the tongue and roof of the mouth. Because TENG can generate triboelectricity based on friction-induced triboelectrification, the various materials can be utilized to fabricate the TENG [27,28,29,30]. Due to the high conversion efficiency from mechanical energy to electrical energy, various TENG-based self-powered sensors have been developed [31,32,33,34,35,36]. Therefore, by adopting the TENG to the motion detection of the tongue, a lightweight and self-powered swallowing rehabilitation monitoring system can be implemented.

Herein, a triboelectric nanogenerator for a swallowing rehabilitation device (TSR) was fabricated. The TSR was operated in the single-electrode mode. The fabricated TSR was composed of lightweight materials including polytetrafluoroethylene (PTFE) and polyethylene terephthalate (PET) to reduce the total weight of the sensor. These materials are materials which can easily be accessed in daily-life and have good biocompatibility [37,38,39,40,41,42,43]. In previous research, reactive ion etching was commonly conducted in the horizontal direction to increase the electrical output generated from the TENG [44,45,46,47,48]. In this study, tilted reactive ion etching (RIE) was conducted with various tilted angles to confirm the effect of the tilted RIE process. The electrical outputs generated from the TSR according to the tilted angle for the tilted RIE process were investigated. The pressure sensitivity was also investigated to confirm the possibility of the TSR as the motion detection for the tongue and pressure sensitivity of 47 mV/kPa and 2.125 nA/kPa were recorded in a low-pressure region (20 kPa to 100 kPa). The potential of the TSR to use in the mouth was demonstrated by checking the tendency of the electrical output when the relative humidity was increased. The surfaces of the PTFE film after conducting the tilted RIE process were investigated through a scanning electron microscope (SEM) by changing the angle of the sample stage and the oblique morphology was successfully observed. The relationship between the oblique morphology and the electrical output generated from the TSR was systematically analyzed and the TSR can detect the pressing, licking, and holding conducted by the tongue. Based on these results, the smart swallowing rehabilitation monitoring system (SSRMS) was implemented as the application. With the proposed SSRMS, weak, medium, and strong pressing by the tongue were successfully detected and classified. The pressed location was detected with the SSRMS. As a result, the proposed SSRMS demonstrated its excellent ability to detect the motion of the tongue. Considering these results, the SSRMS can be expected to be utilized as a promising smart swallowing rehabilitation monitoring system in the near future.

## 2. Materials and Methods

### 2.1. Fabrication of Triboelectric Nanogenerator for Swallowing Rehabilitation (TSR) and Smart Swallowing Rehabilitation Monitoring System (SSRMS)

The reactive ion etching process was performed with O_2_ gas at 30 sccm, at power of 30 W, and time for 180 s on the surface of the commercial PTTE film (active area: 1.5 cm × 2 cm with thickness of 100 μm). To conduct the tilted reactive ion etching, the customized tilted structures were fabricated according to each tilted angle. Then, the PTFE film was attached to the copper electrode as the active layer inducing the triboelectrification with the aid of the double-sided tape. Copper tape possessing only one side of the conductivity was utilized as the electrode. The electrode of the SSRMS was deposited via RF sputtering at 100 W for 30 min. The size of the SSRMS was 3 cm × 3 cm and it was mounted with the aid of the commercial double-sided tape.

### 2.2. Characterization

After the reactive ion etching process, the surface morphology of the PTFE film was investigated using a high-resolution field emission scanning electron microscope (S-4700, Hitachi, Tokyo, Janpan and MERLIN, Carl Zeiss, Jena, Germany). The roughness of the PTFE film was confirmed with an atomic force microscope (XE7, Park Systems, Suwon, Korea).

### 2.3. Electrical Measurement

The electrodynamic shaker (LW139.138-40, Labworks Inc., Costa Mesa, CA, USA) was utilized to apply a quantitative force to the TSR and the applied force was measured through a force sensor (1053V4, Cylos, Suwon, Republic of Korea). The pressure was calculated by dividing the force by the area. The relative humidity was measured by a thermo-hygrometer. A commercial humidifier was utilized to increase the relative humidity and an acrylic case was utilized to maintain the relative humidity. An electrometer (Keithley 6514, Tektronix, Beaverton, OR, USA) was utilized to measure the open-circuit voltage (*V*_OC_) and short-circuit current (*I*_SC_). The Arduino nano 33 BLEs were utilized to measure the output voltage generated from the smart swallowing rehabilitation monitoring system (SSRMS).

### 2.4. Implementation of the Smart Swallowing Rehabilitation Monitoring System

To implement the smart swallowing rehabilitation monitoring system, the Python environment was utilized through Project Jupyter.

## 3. Results and Discussions

### 3.1. Fabricate the Triboelectric Nanogenerator for Swallowing Rehabilitation

Figure 1a shows the image diagram of the fabricated triboelectric nanogenerator for a swallowing rehabilitation (TSR). The TSR was composed of polyethylene terephthalate (PET) as a top layer substrate, a copper (Cu) electrode, and polytetrafluoroethylene (PTFE) as a triboelectric inducing layer. PTFE and PET films were utilized due to the biocompatibility of the materials. The PTFE film was utilized to fabricate the proposed TSR after conducting reactive ion etching (RIE) on its surface in order to form the oblique morphology at the PTFE. The surfaces of the PTFE before and after the RIE process were revealed at Figure 1b. Before conducting the RIE process, the specific oblique morphology was not observed on the surface of the PTFE film. However, after conducting the RIE process, the oblique morphology was formed on the PTFE film. The proposed TSR was designed as the single-electrode mode triboelectric nanogenerator (TENG). This is because the single-electrode mode TENG is suitable for use in the mouth due to the advantage of its simple structure for generating triboelectricity compared to other operation modes of the TENG. The working mechanism of the fabricated TSR is illustrated on Figure 1c. The electrons are easily attracted to the PTFE after rubbing between the PTFE film and external material because of the high electronegativity of the PTFE film owing to the large amount of fluorine. The negative charges on the external material do not dissipate easily if the conductivity of the external material is not high. When the external material is detached from the surface of the PTFE film, the effect of the negative charges on the external material is decreased and the electrical equilibrium is broken. The broken electrical equilibrium induces the stream of the electron from ground to Cu electrode because the electron can easily be moved in the electrode compared to the insulator. When the external material fully separates from the TSR, the effect of the negative charges is removed and the electrical equilibrium is balanced by the Cu electrode and PTFE film. Then, as the external material starts to move toward the PTFE film, the electrons in the Cu electrode begin to be pushed away from the electrode to ground due to the negative charges on the external material. As a result, AC electricity is generated from the TSR.

### 3.2. Electrical Output Generated from the TSR

The open-circuit voltage (*V*_OC_) and short-circuit current (*I*_SC_) generated from the TSR are presented in Figure 2a and b, respectively. The *V*_OC_ and *I*_SC_ generated from the TSR with the bare PTFE film (TSR-B) were recorded as 13.10 V and 0.512 μA, respectively. On the other hand, the TSR with the PTFE film conducting the tilted RIE for 60° (TSR-T60) generated *V*_OC_ of 24.12 V and *I*_SC_ of 1.056 μA. Compared to the *V*_OC_ and *I*_SC_ from the TSR-B, those from the TSR-T60 were dramatically increased by 184% and 206%, respectively. This dramatic rise of electrical output was due to the contribution of the oblique morphology on the TSR-T60. After conducting the tilted RIE process, the oblique morphology was formed on the PTFE film of the TSR. As a result, the effective contact area of the TSR increased due to the oblique morphology. Because the amount of the generated triboelectricity is determined by the contact area between two different materials, the electrical output generated from the TSR-T60 should be higher than that from the TSR-B. The nomenclature of the abbreviations can be checked in the Table 1.

In Figure 2c, the electrical outputs generated from the TSR-B and TSR according to the tilted angle for the RIE process were investigated. The tilted RIE process was conducted as shown in diagram of Figure 2c and the tilted angle was decided based on the bottom line of the reactive ion etcher and the bevel face of the customized structure. The tilted angle for the RIE process was changed from 0° to 75° with increments of 15°. Each mean value of generated electrical output and its standard deviation can be checked in Table 2. As a result, the optimal tilted angle to increase the electrical output generated from the TSR was recorded as 60° as shown in Table 2. The electrical output generated from the TSR after conducting the tilted RIE at the angle of 50°, 55°, 60°, 65°, and 70° can be checked in Appendix A.

To utilize the TSR as the swallowing rehabilitation monitoring system, the pressure sensitivity of the TSR in the low-pressure region should be considered as the important factor because the pressure applied by the tongue to the TSR will be not intensive. Hence, the pressure sensitivity of the TSR was investigated with the range of 20 kPa to 3400 kPa as shown in Figure 2d. Figure 2d is divided into region 1 with weak pressure (R1) and region 2 with strong pressure (R2). Pressures ranging from 20 kPa to 180 kPa were allocated to R1. On the other hand, pressures from 300 kPa to 3400 kPa were assigned to the R2. Each electrical output generated from the TSR-B and TSR-T60 according to the pressure can be checked in Table 3. In R1, the slopes of 35 and 47 mV/kPa were calculated with TSR-B and TSR-T60, respectively. In R2, values of *V*_OC_ generated from TSR-B and TSR-T60 increased by 1.10 and 2.20 mV/kPa, respectively. In terms of the current, 3.70 times and 1.15 times greater values were observed in the R1 and R2 of the TSR-B and TSR-T60, as shown in Appendix A. After the tilted RIE process, the surface roughness of the PTFE film was increased, indicating that the effective area of the PTFE film was increased. With a bare and tilted angle of 30°, and 60°, each roughness of the PTFE film was 78 nm, 90 nm, and 95.8 nm, respectively, as shown in Appendix A. Their results indicated that the empty space between the surface morphology of the PTFE increased as the tilted angle increased. Hence, with the increment of pressure, the possibility of the full contact with this empty space increased, inducing the increasing electrical output. Considering these, the tilted RIE process had a greater impact on the increment of the electrical output of the TSR than the perpendicular RIE process. Considering these results, the high-pressure sensitivity of the TSR-T60 was successfully demonstrated. This result indicates that the possibility of the TSR-T60 as the detection sensor for the tongue movement is confirmed.

It is important to confirm the stability of the electrical output generated from the TSR-T60 because the inside of the mouth possesses high humidity conditions compared to the outside of the mouth. Therefore, the electrical output generated from TSR-T60 was investigated when the relative humidity was changed from 30% to 80% in increments of 10%. The relative value was employed to easily compare the electrical output according to the relative humidity. The electrical output generated from each TSR was divided based on the value of the *V*_OC_ generated from the TSR-B acquired at the relative humidity of 30%. A decrement of 12.82% in the electrical output was observed with the TSR-B when the relative humidity was 80%. The electrical output generated from the TSR with the PTFE film after conducting the tilted RIE with the tilted angle of 30° (TSR-T30) was decreased 10.68% at the relative humidity of 80% compared to the generated electrical output from the relative humidity of 30%. The smallest decrement in the generated electrical output was observed with the TSR-T60 when the relative humidity was changed from 30% to 80%. The electrical output of the TSR-T60 decreased only 8.80% in the relative humidity of 80% compared to that of the TSR-T60 in relative humidity of 30%. This result indicates that the fabricated TSR-T60 has high resistance to humidity and that it can be utilized inside of the mouth. After the tilted RIE process, the increased hydrophobicity of the PTFE film was confirmed, as shown in Appendix A. These results indicate that water can be easily removed from the surface of the TSR and, therefore, the effect of water on the electrical output generated from the TSR was small. Additionally, the electrical power density was calculated to confirm the optimal load resistance in order to maximally transfer the electrical power to the sensor. As a result, power density of 122.49 mW/m^2^ was observed at 20 MΩ with the TSR-T60, as shown in Figure 2f. The generated electrical output voltage and current can be checked in Appendix A.

### 3.3. Surface of the PTFE Film after Conducting the Tilted Reactive Ion Etching Observed by the Scanning Electron Microscope

In Figure 3, surface images of the bare PTFE film (Figure 3a) and PTFE films after RIE process with the tilted angle of 0° (Figure 3b) and 60° (Figure 3 3c) were revealed by SEM with the sample stage tilted at 0° and 15°. Also, surface images of the PTFE films after tilted RIE process in Appendix A (tilted angle for RIE process: 0°, 15°, and 30°) and Appendix A (tilted angle for RIE process: 45°, 60°, and 75°) by SEM with the sample stage tilted at 0°, 15°, and 30°, respectively.The tilted RIE process was conducted by changing the tilted angle from 0° to 75° with an increment of 15°. The plasma for etching was vertically aligned with the bottom line of the reactive ion etcher, the shape of the morphology through the etching should follow the colliding direction between the plasma and PTFE film. Hence, when the tilted RIE process is conducted, the incident angle of the plasma is changed and the etched surface is also changed compared to the RIE process with the tilted angle of 0° because the inclined plasma is etched on the surface. Compared to the bare PTFE film, the surface images of the PTFE films after conducting the tilted RIE process (0°, 15°, and 30°) showed the height in the oblique morphology. The height can be distinguished based on the brightness of the oblique morphology displayed at the SEM images. The surface images of the PTFE films after conducting the tilted RIE process were also investigated by tilting the sample stage of the scanning electron microscope to 15° and 30° to confirm the oblique morphology on the PTFE film. Especially, the oblique morphology formed on the surface of the PTFE film after conducting the tilted RIE at the angle of 30° can be easily observed when the sample stage of the SEM was tilted at 15°. When the tilted angle for the RIE process exceeds 30°, the oblique morphology becomes difficult to notice by tilting the sample stage of the SEM at angles of 0° and 15°, as shown in Appendix A. However, the oblique morphology formed on the PTFE films after conducting the tilted RIE process, which exceeds the tilted angle of 30°, was easily observed when the sample stage of the SEM was tilted to 30°. When the tilted angle was 75°, the oblique morphology began to decrease. The strength of C–C in the PTFE is 348 kJ/mol (57.78 × 10^−20^ N/m). The length of the C–C bond is 154 pm. Therefore, the force between C–C bond can be calculated as 3.752 nN. When the surface structure is vertical, it only needs to withstand gravity in order to maintain the standing state. However, when the surface structure is inclined, it needs to withstand gravity and additional torque (sin(θ) × 3.752 nN) in order to maintain the standing state. Therefore, the larger tilted angle induces a larger additional torque to the structure. As a result, the structure with a large tilted angle easily falls down, as shown in Appendix A. This is also consistent with the results in Figure 2c. The collapsed oblique morphology decreased the effective contact area for inducing the triboelectricity and the electrical output generated from the after conducting the tilted RIE at the tilted angle of 75° was decreased. On the other hand, the proper tilted angle for the RIE process can create the oblique morphology on the surface of the PTFE film and the effective contact area can be increased because of the oblique morphology, where the surface of the PTFE is in contact with the external material. This result is also evidence for the trend of the electrical output in Figure 2c.

### 3.4. Durability of the TSR and Ability to Detect the Motion of the Tongue with the TSR

The long-term durability of the TSR-T60 was investigated to confirm the stability of the oblique morphology as shown in Figure 4a. The approximately *V*_OC_ of 28 V was generated during 10,800 cycles without any specific degradation. This result indicated that the fabricated TSR-T60 can maintain its initial electrical performance despite being used. Based on this result, the ability to distinguish the motions of the tongue was confirmed with the TSR-T60. When the tongue was weakly in contact with the TSR-T60, *V*_OC_ of 0.82 V was generated. *V*_OC_ of 6 V was generated from the TSR-T60 with normal contact between the tongue and TSR-T60. Finally, when the tongue was strongly in contact with the TSR-T60, *V*_OC_ of 9.24 V was generated from the TSR-T60. These results are shown in Figure 4b. When the tongue was in contact with the TSR-T60, the generated voltage was dramatically increased. On the other hand, when the tongue was detached from the TSR-T60, the voltage was not decreased in a short time. This is because the body fluids remain on the surface of the TSR-T60 after contact. Rehabilitation exercises such as sliding the tongue against the roof of the mouth or continuously tapping the tongue against the roof of the mouth are usually carried out to rehabilitate the swallowing function. Therefore, the motion of the tongue should be detected with the proposed TSR-T60 to conduct the swallowing rehabilitation monitoring. The three motions including the pressing, licking, and holding with the tongue were applied to the fabricated TSR-T60 and the generated electrical outputs were investigated as shown in Figure 4c. Each motion of the exercise was described in Figure 4c. As a result, the *V*_OC_ of 8.21 V, 2.12 V, and 1.54 V were generated from the TSR-T60 with the exercise of the pressing, licking, and holding by the tongue, respectively. The highest electrical output was generated from the pressing exercise because pressing the roof of the mouth with the tongue was the most powerful exercise to be performed. Because the tongue possesses a wet surface and relatively high conductivity, the discharge of triboelectric charge can occur. To suppress this, we designed the TSR to prevent the direct contact between Cu electrode and tongue by attaching the PTFE film to the Cu. The PTFE possesses hydrophobic surface and high electronegativity. Saliva consists mainly of water, which can be easily removed from the hydrophobic surface of the PTFE film. Hence, the PTFE film can possess high resistance to wet environment as shown in Figure 2e. As a result, saliva can be easily removed from the PTFE film due to its hydrophobic surface. Thus, the TSR can generate stable electrical output in the mouth without specific discharge. Based on these results, the possibility of the fabricated TSR-T60 as the sensor for detecting tongue motion is successfully demonstrated.

### 3.5. Implementation of the Smart Swallowing Rehabilitation Monitoring System

Considering the advantages of the TSR-T60 to detect the motion of the tongue, the smart swallowing rehabilitation monitoring system (SSRMS) was implemented as shown in Figure 5a. The SSRMS consisted of a sensor part to acquire the data from the motion of the tongue, a central part to transfer the acquired data through the wireless communication, a peripheral part to receive the transferred data, and a main server to process the received data. The TSR-T60 array was utilized as the sensor to acquire the data from the motion of the tongue. The TSR-T60 sensor array consisted of three channels with the compact size of 3 cm × 3 cm to insert into the inside of the mouth. As the devices for the central and peripheral, the two Arduino nano 33 BLEs were utilized. The generated signal from the TSR-T60 array was also acquired by the Arduino nano 33 BLE by in parallel connecting the resistance of the 1 GΩ. Then, the generated electrical signal was transferred from the central part to the peripheral part through the embedded BLE module in the Arduino nano 33 BLE. Then, the transferred data was processed in the main server by the customized software at the Python environment. The acquired data from the TSR-T60 array are shown in Figure 5b. The policy to determine the intensity of the pressure based on the output voltage was created by observing the electrical output when the tongue was in contact with the TSR-T60 array. To suppress the ghosting signal due to the switching of the multiplexer embedded in the Arduino nano 33 BLE, the 20 ms of the time delay was adopted to discharge the capacitor in the multiplexer [49,50]. The range under the 2.0 V was classified as the weak pressure and the range within 2.0 V to 2.5 V was assigned as the middle pressure and the range above the 2.5 V was allocated to the strong pressure. At the first time, the tongue was in contact with the channel 1 of the TSR-T60 array and the output voltage of 2.5 V was generated. The touch event in channel 1 was successfully detected with the proposed SSRMS and the intensity of the pressure was also detected simultaneously. Compared to the first time, lower pressure was applied to channel 2 of the TSR-T60 array and 1.8 V was generated as the output voltage from channel 2 of the TSR-T60. The pressure intensity and the touched channel were successfully detected with the SSRMS. When the pressure was applied to channel 3 of the TSR-T60 array, the highest output voltage of 2.8 V was generated and it could be observed in the monitor of the SSRMS. Considering these results, the proposed SSRMS can be expected to be utilized as a smart swallowing rehabilitation monitoring system in near future because the SSRMS successfully demonstrated its great potential in terms of motion detection of the tongue.

## 4. Conclusions

In summary, a triboelectric nanogenerator for swallowing rehabilitation (TSR) was fabricated. Compared to the bare TSR, the TSR after conducting the tilted RIE process at the angle of 60° (TSR-T60) generated an open-circuit voltage (*V*_OC_) of 24.12 V and short-circuit current (*I*_SC_) of 1.056 μA, increases of 189% and 237%, respectively. The electrical outputs generated from the TSR were measured according to the tilted angle for the tilted RIE process and the angle of 60° was the optimal angle to increase the electrical output generated from the TSR. The pressure sensitivity was also investigated and pressure sensitivities in the low-pressure region (20 kPa to 100 kPa) of 47 mV/kPa and 2.125 nA/kPa were recorded. Electrical output according to the relative humidity was checked to confirm the possibility for using the TSR inside the mouth and a decrement of only 8.80% in electrical output was observed. The effect of the tilted RIE process on the electrical output generated from the TSR was systematically analyzed. Additionally, the oblique morphology was investigated with the scanning electron microscope (SEM) by changing the angle of the sample stage and the oblique morphology were successfully observed. The pressing, licking, and holding motions of the tongue were detected with the TSR-T60. Based on these results, the smart swallowing rehabilitation monitoring system (SSRMS) was implemented as the application. The proposed SSRMS was composed of a sensor part, central part, peripheral part, and main server, respectively. With the proposed SSRMS, pressing of various intensities was successfully detected and so was the location where the contact between tongue and TSR-T60 based sensor array occurred. As a result, the proposed SSRMS demonstrated its excellent ability to detect the motion of the tongue. Hence, the SSRMS can be expected to be utilized as a smart swallowing rehabilitation monitoring system in near future.

## Figures and Tables

**Figure 1 materials-15-02240-f001:**
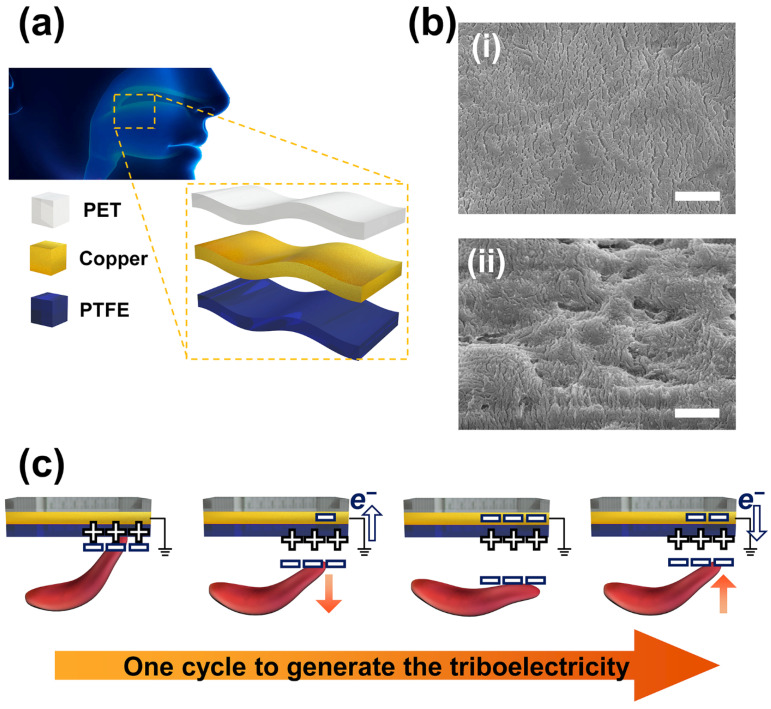
Diagram of the fabricated TSR. (**a**) The schematic image of the TSR. (**b**) The surface of the PTFE film observed by the SEM with scale bar indicating 1 μm. (**i**): Surface of bare PTFE. (**ii**) Surface of tilted PTFE. (**c**) The working mechanism to generate triboelectricity for one cycle with the TSR.

**Figure 2 materials-15-02240-f002:**
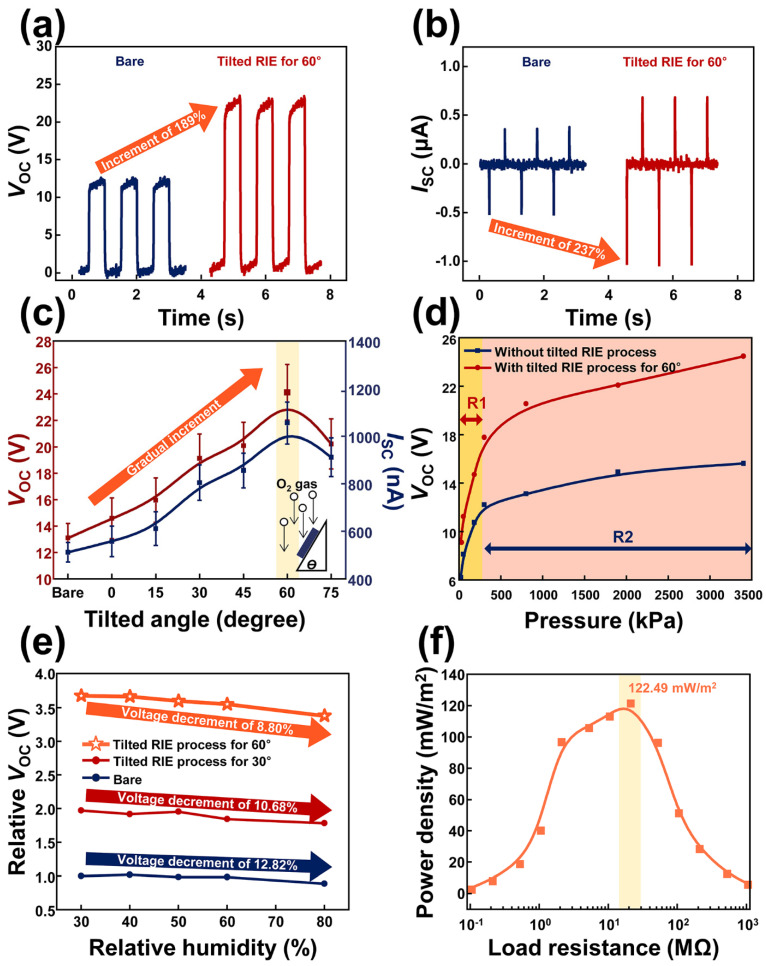
Electrical output generated from the TSR. (**a**) Open−circuit voltage and (**b**) short−circuit current generated from the TSR-B and TSR-T60. (**c**) Electrical output generated from the TSR according to the tilted angle. (**d**) Pressure sensitivity of the TSR−B and TSR−T60. (**e**) The relative open−circuit voltage according to the relative humidity. (**f**) Output power density with the various load resistances.

**Figure 3 materials-15-02240-f003:**
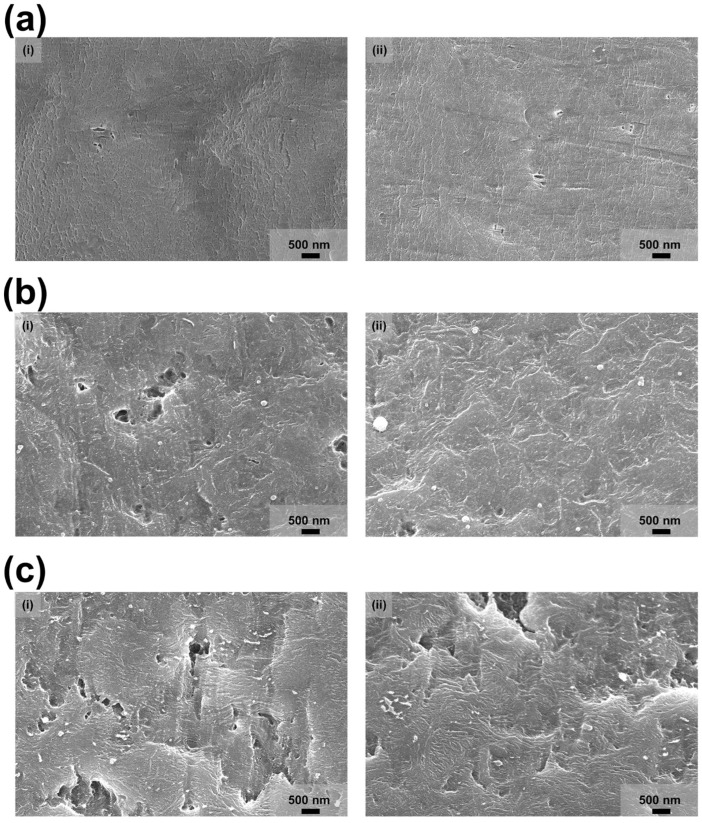
The surface of the PTFE film observed by the scanning electron microscope. (**a**) The surface of the bare PTFE film observed by the SEM with the tilted angle of the sample stage of (**i**) 0° and (**ii**) 15°. (**b**)The surface of the PTFE film after conducting the tilted reactive ion etching with the tilted angle of 0° by the SEM with the tilted angle of the sample stage of (**i**) 0° and (**ii**) 15°. (**c**) The surface of the PTFE film after tilted reactive ion etching with the tilted angle of 60° observed by the SEM with the tilted angle of the sample stage of (**i**) 0° and (**ii**) 15°, respectively.

**Figure 4 materials-15-02240-f004:**
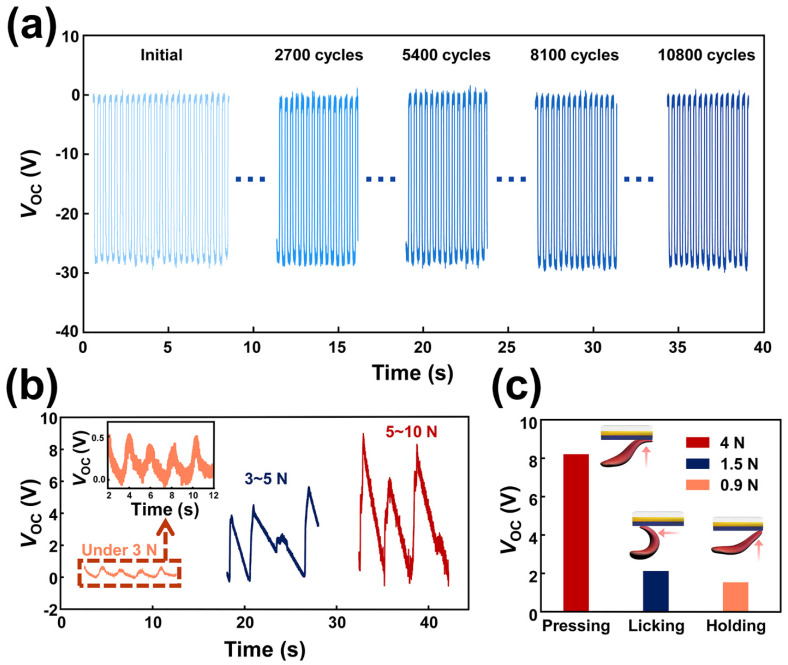
The durability and ability to detect the motion of the tongue. (**a**) The durability of the fabricated TSR−T60 with the working frequency of 3 Hz. (**b**) Electrical output generated from the TSR−T60 by weak (orange), middle (blue), and strong (red) pressing by the tongue. (**c**) Electrical output generated from the TSR−T60 by pressing, licking, and holding of the tongue with a force of 4 N, 1.5 N, and 0.9 N.

**Figure 5 materials-15-02240-f005:**
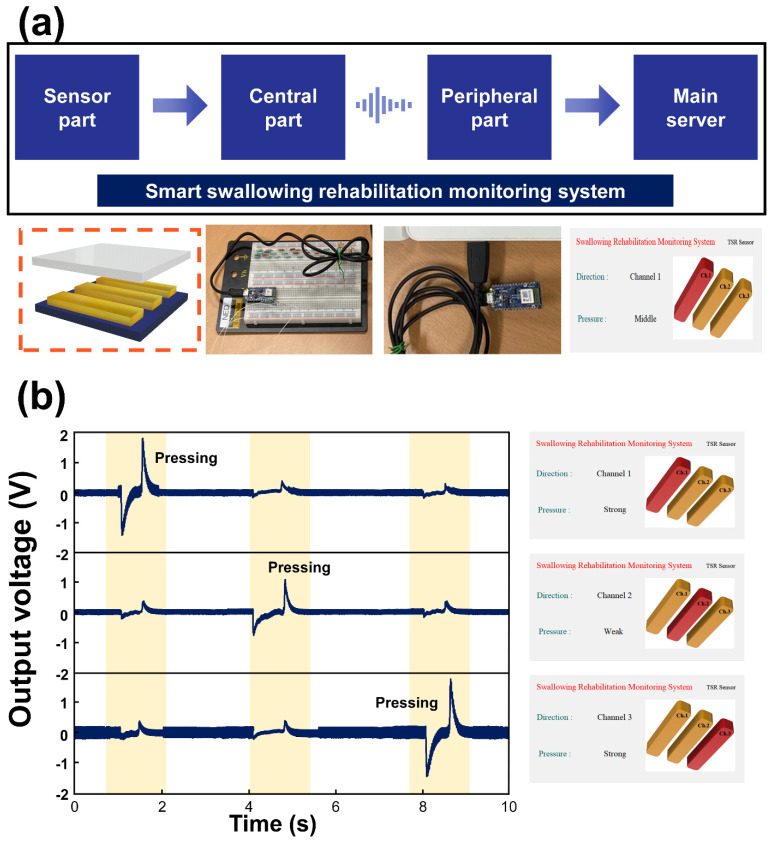
The smart swallowing rehabilitation monitoring system. (**a**) The schematic of the proposed smart swallowing rehabilitation monitoring system (SSRMS) and its components. (**b**) The electrical output generated from the SSRMS and the detecting results displayed on the monitor of the customized software.

**Table 1 materials-15-02240-t001:** Nomenclature of the abbreviations.

Nomenclature			
Abbreviations		Abbreviations	
*V* _OC_	Open-circuit voltage	TSR	TENG for the swallowing rehabilitation
*I* _SC_	Short-circuit current	TSR-B	TSR with bare PTFE film
TENG	Triboelectric nanogenerator	TSR-T60	TSR with PTFE film conducting the tilted RIE for 60°
PTFE	Polytetrafluoroethylene	SSRMS	Smart swallowing rehabilitation monitoring system
PET	Polyethylene terephthalate	SEM	Scanning electron microscopy

**Table 2 materials-15-02240-t002:** Each mean value of electrical output generated from the TSR and its standard deviation according to the tilted angle.

Tilted Angle	Bare	0°	15°	30°	45°	60°	75°
Mean value of *V*_OC_ (V)	13.10	14.64	16.08	19.21	20.10	24.12	20.22
Standard deviation	1.1	1.54	16.8	1.82	1.75	2.1	1.89
Mean value of *I*_SC_ (nA)	512	558	612	805	856	1056	912
Standard deviation	41	64	71	75	73	88	81

**Table 3 materials-15-02240-t003:** Each electrical output generated from TSR-B and TSR-T60 according to the applied pressure.

	Pressure (kPa)	20	50	180	300	800	1900	3400
Without tilted RIE process	*V*_OC_ (V)	6.21	8.14	11.77	12.22	13.13	14.94	15.64
With tilted RIE process for 60°	*V*_OC_ (V)	9.11	11.25	14.72	17.79	20.56	22.09	24.49
Without tilted RIE process	*I*_SC_ (nA)	112	131	202	276	332	529	676
With tilted RIE process for 60°	*I*_SC_ (nA)	152	205	407	503	749	866	1011

## Data Availability

Not applicable.

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
