# Peer review of "Self-Powered and Flexible Triboelectric Sensors with Oblique Morphology towards Smart Swallowing Rehabilitation Monitoring System"

_materials, 2022, doi:10.3390/ma15062240_

Round 1

Reviewer 1 Report

This paper provides a self-powered flexible triboelectric nanogenerator with tilted nanostructure towards smart swallowing rehabilitation monitoring system. The author studied the influence of tilted reactive ion etching on electrical output and pressure sensitivity. This study is systematic and detailed, but I think the output performance of this device is general and lack of novelty. Therefore, the manuscript cannot meet the academic level of Materials.

  1. For the triboelectric nanogenerator, PTFE has been widely used as the negative material. Hence, using PTFE as the triboelectric material is common and lacks novelty.
  2. The author made the nanostructure on the film surface by ion reactive etching, which has been studied in previous studies (Advanced Materials, 29(2017), 1703456). Hence, the fabrication process is not innovative.
  3. The open circuit voltage of the developed device is only 23.89 V, which is lower than the several previous studies (Nano Research 2016, 9(5): 1442–1451).
  4. Considering the safety of the material, according to the list of carcinogens of the international agency for research on cancer of the World Health Organization, polytetrafluoroethylene is a Class 3 carcinogen; hence, the material is not suitable for wearable devices to monitor swallowing.

Author Response

Dear respect Reviewer,

We are sincerely grateful to the editor and the reviewers for all the valuable comments and advice. Revisions have been carefully conducted with our best effort and point-by-point responses are given as follows. The sentences outside of the rectangular boxes are the comments from the reviewer and the sentences inside of the rectangular boxes are our responses to the reviewers’ comments.

Please kindly check the attached file for response.

Warm regards,

Reviewer 2 Report

This manuscript demonstrates a self-powered and flexible triboelectric sensor to monitor smart swallowing rehabilitation. It is well written and can be accepted after minor revision.

The length of scale bars in Figure 3 should be given.

Author Response

Dear respect reviewer,

We are sincerely grateful to the editor and the reviewers for all the valuable comments and advice. Revisions have been carefully conducted with our best effort and point-by-point responses are given as follows. The sentences outside of the rectangular boxes are the comments from the reviewer and the sentences inside of the rectangular boxes are our responses to the reviewers’ comments.

Warm regards,

Reviewer 3 Report

In this manuscript (materials-1562675) the authors illustrated “Self-powered and flexible triboelectric sensors with the tilted nanostructures towards smart swallowing rehabilitation monitoring system”. A major revision is required to enhance the quality of manuscript and to effectively elaborate the given findings.

  1. Abstract of the manuscript should be re-write?
  2. Introduction section needs to be enhanced by adding following references?

Journal of King Saud University – Science 32, 4, 2020, 2397-2405;

 https://doi.org/10.1016/j.est.2020.102125;

  1. The pressure sensitivity is increased by 237% and 432%, it this the maximum value reported so far?
  2. There is no scale bar on SEM images, use the high-quality scale markers and good SEM images.
  3. The results and discussion sections need an improvement.
  4. English of the manuscript needs to polished and rectified thoroughly.

Author Response

(The authors gave the same response as above.)

Reviewer 4 Report

The paper has many open questions.

  1. What is the reason to use the PET?
  2. One part of the TENG has a watery surface and is highly conductive (tongue). How is the discharge of charge through the tongue avoided? 
  3. How can triboelectricity be avoided?
  4. Which gas was used for the RIE-process?
  5. What power density was applied to the samples in the plasma prozess?
  6. Why is the structural formation of the surfaces of perpendicularly incident ions less than that of inclined ones?
  7. Why should inclined ions have a greater impact than perpendicular ones? 
  8. Which gas etched the surface?
  9. At what temperature was the RIE process performed?  
  10. How big were the samples used for the measurements? 
  11. For what reason were the specified pressure values selected?
  12. What explains the influence of pressure on static electricity?
  13. How are the single channels connected in the SSRMS?
  14. Why do the channels deliver signals if the are not excited?

Figure 3 shows several SEM-pictures. Unfortunately, it is impossible to draw conclusions from the structures at different ion incidence angles from these images. The image sections are much too small! The designation of Figs. is inconsistent. Sometimes there are indications like S4 (line 288).

Sometimes words are missing or shifted (lines 75, 78) 

Who would use a system with dimensions of 5cm x5cm inside the mouth?

Author Response

(The authors gave the same response as above.)

Reviewer 5 Report

The paper presents results on the formation and study of triboelectric sensors. The approach is traditional utilizing a combination of metal film with polymer layers. The originality is in use of reactive ion etching (RIE) at different angles allowing to change the polymer morphology and get enhancement of the electrical signal. The paper can be recommended to publication but only after major revision. There is a number of points which need to be clarified and explained properly. The following issues should be considered by the authors.

  1. Last section of Introduction, lines 75-104, looks like an extended abstract. It can be shortened just to a few sentences emphasising originally and main findings of the paper.
  2. As follows from the paper, the authors modify the polymer surface by RIE applying the beam at different angles to the surface. This, however, does not mean that the etching leads to the formation of tilted nanostructures. One can not see anything like that in the SEM images presented in the article. Etching at different impact angles just changes the surface morphology. Thus, term “tilted nanostructures” should be removed from the title and substituted by other proper term/phrase in the text because it is misleading for the readers.
  3. Section 2 requires more detailed description of all production and characterisation steps.
  • For the REI regimes: which ions are used, at which angles etc. Some of this information can be found at different places in the text but it would be much better to describe it properly in the methods.
  • Also, more details about sensors manufacturing are required; how polymer films are made and deposited, how copper electrodes are produced?
  • How electrical contacts were made to do the electrical measurements? How the voltage measurements for different pressure and humidity values were carried out?
  • What does it mean weak, middle and strong pressing the tongue, can it be quantified? How the measurements inside of a mouth were performed? How the relatively big sensor (5x5 cm2) was mounted in a mouth?
  • The section should also list all the samples produced explaining the abbreviations (only once), which are then used in the text. Information on how many samples of the same type were studied is missing. Without this information, precise declaration of increased efficiency (by 237% or 432%) of the REI treated samples compared to untreated ones does not make sense. What are standard deviations in electrical parameters of your samples?
  1. In section 3.3, there are paragraphs where the authors list voltage and current numbers for different angles, different pressures etc., thus, producing long hardly readable sentences. First, it is a kind of repletion of the information presented in figure 2. Second, if the authors want these details, it would be better to organise them into tables. This section also misses any explanations or at least hypothesis why the electrical output changes depending on the treatment REI angle, humidity, pressure etc.
  2. In section 3.3, there is a statement “the shape of the formed nanostructures through the etching should be following the colliding direction between the plasma and PTFE film”. Can this statement be grounded by literature or better explanation of physics? SEM images do not show any tilted nanostructures. There is only change of the surface morphology due to the difference in geometry of the ion beam interacting with the surface. From this point of view, schematic presented in Figure S5 has no experimental confirmation, i.e. pure speculation. Also, the statement that “height can be distinguished based on the brightness of the nanostructures displayed at the SEM images” is wrong. The contrast can be related to change of morphology, structure and composition. Height difference can not be distinguished unless you provide some reference/calibration measurements.

The assumption in lines 259-260 that “the tilted nanostructures cannot withstand gravity” has no sense. The authors should consider the strength of chemical bonds in polymers and compare with possible gravitational forces acting on small protrusions (simple calculation).

Minor comments.

The paper should go through proof reading to remove a number of typos. Some sentences seem to be not completed or have strange grammatical structure.

In panel b of figure 1 it should be clearly stated that (i) is untreated while (ii) is treated sample. Better linking of panel c and text is required.

In figure 3, scale for a bar is missing. Only by comparing with figure 1, one can assume that it is 1 micron. Actually, to address formation of any nanostructures, higher SEM magnification would be appreciated.

What are the numbers at top of panel a in figure 4? In panel c of the same figure, some values in N are presented. They need to be explained. Also, see my comment to section 2.

Author Response

(The authors gave the same response as above.)

Round 2

Reviewer 1 Report

The manuscript has been improved and I support its publication in its current form.

Author Response

Dear respect reviewer,

We would appreciate for reviewing our manuscript.
Based on reviewer`s comment, the quality of our manuscript can be enhanced.

Warm regards,

Reviewer 4 Report

I still have fundamental doubts about the explanations of the function of the sensor and the supposed effect of the inclination of the samples in the RIE process 

Author Response

Dear respect reviewer,

We are sincerely grateful to the editor and the reviewers for all the valuable comments and advice again. Revisions have been carefully conducted with our best effort and point-by-point responses are given as follows. The sentences outside of the rectangular boxes are the comments from the reviewer and the sentences inside of the rectangular boxes are our responses to the reviewers’ comments.

Warm regards,

Reviewer 5 Report

The revised version is considerably improved and can be recommended for publication.

Author Response

(The authors gave the same response as above.)
